Early Miocene remains of Melissiodon from Mokrá-Quarry (Moravia, Czech Republic) shed light on the evolutionary history of the rare cricetid genus

Bonilla-Salomón Isaac salomon1@uniba.sk 1
Čermák Stanislav 2
Luján Àngel H. angel.lujan@icp.cat 3 4
Jovells-Vaqué Sílvia 1 4
Ivanov Martin 3
Sabol Martin 1
1 Department of Geology and Paleontology, Comenius University in Bratislava , Bratislava , Slovak Republic
2 Institute of Geology of the Czech Academy of Sciences , Prague , Czech Republic
3 Department of Geological Sciences, Faculty of Science, Masaryk University , Brno , Czech Republic
4 Institut Català de Paleontologia Miquel Crusafont, Universitat Autònoma de Barcelona , Cerdanyola del Vallès , Spain
López-Antoñanzas Raquel
Electronic publication date: 2022 Aug 8
Publication date: 2022
Volume: 10
Electronic Location ID: e13820
Received 2022 Feb 16; Accepted 2022 Jul 8
Copyright: ©2022 Bonilla-Salomón et al.
Copyright year: 2022
Copyright holder: Bonilla-Salomón et al.
License: This is an open access article distributed under the terms of the Creative Commons Attribution License, which permits unrestricted use, distribution, reproduction and adaptation in any medium and for any purpose provided that it is properly attributed. For attribution, the original author(s), title, publication source (PeerJ) and either DOI or URL of the article must be cited.
License URL: https://creativecommons.org/licenses/by/4.0/

Keywords: Melissiodontinae, Melissiodon schlosseri, Melissiodon dominans, Burdigalian, Carpathian Foredeep Basin, Moravian Karst

Funding: The VEGA Scientific Agency 1/0164/19, 1/0533/21 The APVV grant agency of the Slovak Republic APVV-16-0121 APVV-20-0079 The UK grant of the Comenius University in Bratislava UK/56/2019 UK/100/2020 UK/75/2021 The Institute of Geology of the Czech Academy of Sciences RVO67985831 The NSP (National Scholarship program of Slovak Republic for the Support of Mobility of University Researchers) 33758 The Specific research project at the Faculty of Science at Masaryk University MUNI/1263/2020 The Ministerio de Ciencia e Innovación and the Agencia Estatal de Investigación I+D+I PID2020-117289GB-I00 funded by MCIN/AEI/10.13039/501100011033/ The Operational Programme Research, Development and Education –Project ‘Postdoc@MUNI’ No. CZ.02.2.69/0.0/0.0/16_027/0008360 The Generalitat de Catalunya (CERCA Program and 2017 SGR 116) The ‘Programa Postdoctoral Beatriu de Pinós de la Secretaria d’Universitats i Recerca del Departament d’Empresa i Coneixement de la Generalitat de Catalunya’ (2019 BP 00154) The research reported in this work has been funded by: the VEGA Scientific Agency (Project No. 1/0164/19; Isaac Bonilla-Salomón and; Martin Sabol; and Project No. 1/0533/21; Martin Sabol); the APVV grant agency of the Slovak Republic (Project No. APVV-16-0121 and APVV-20-0079; Isaac Bonilla-Salomón; Martin Sabol); the UK grant of the Comenius University in Bratislava (Project No. UK/56/2019, UK/100/2020 and UK/75/2021; Isaac Bonilla-Salomón); the institutional support RVO67985831 of the Institute of Geology of the Czech Academy of Sciences (Stanislav Čermák); the NSP (National Scholarship program of Slovak Republic for the Support of Mobility of University Researchers (ID 33758 to Sílvia Jovells-Vaqué); the Specific research project at the Faculty of Science at Masaryk University, Brno (MUNI/1263/2020; Martin Ivanov). Àngel H. Luján also received funding from the Ministerio de Ciencia e Innovación and the Agencia Estatal de Investigación (Grant I+D+I PID2020-117289GB-I00 funded by MCIN/AEI/10.13039/501100011033/), the Operational Programme Research, Development and Education –Project ‘Postdoc@MUNI’ (No. CZ.02.2.69/0.0/0.0/16_027/0008360), the Generalitat de Catalunya (CERCA Program and 2017 SGR 116) and the ‘Programa Postdoctoral Beatriu de Pinós de la Secretaria d’Universitats i Recerca del Departament d’Empresa i Coneixement de la Generalitat de Catalunya’ (2019 BP 00154). The funders had no role in study design, data collection and analysis, decision to publish, or preparation of the manuscript.

==============================
Background

Melissiodon is a rare cricetid genus endemic to Europe, known from the Early Oligoceneto the Early Miocene. It is usually a very rare find, and even in the few localities where Melissiodon remains are found, those are scarce and fragmentary. Only a few Central European localities have yielded rich remains of the genus. Currently, two species are known from the Early Miocene: Melissiodon schlosseri, which is based on two teeth from the MN2 German locality of Haslach and only found in two other sites of similar age (Ulm-Uniklinik and La Chaux, from Germany and Switzerland respectively); and Melissiodon dominans, found in MN3 and MN4 localities across Europe, even though the scarce and fragmentary remains make some of these attributions dubious. For that reason, Melissiodon dominans has become a catch-all species. However, Mokrá-Quarry represents one of the best documented findings of Melissiodon remains from MN4 localities of Europe.

Methods

The Melissiodon assemblage from Mokrá-Quarry has been studied thoroughly, providing metrics and detailed descriptions of all teeth positions, as well as complete comparisons with other MN3 and MN4 localities bearing Melissiodon remains.

Results

In this work, new remains of Melissiodon have been identified as a new morphotype that clearly differs from Melissiodon dominans by its unique m1 morphology but still shows some resemblance with Melissiodon schlosseri. Based on that, we here propose the hypothesis of an evolutionary lineage starting from Melissiodon schlosseri, diverging from the lineage leading towards Melissiodon dominans. With this finding, there are at least two different taxa of Melissiodon known during the Early Miocene, prior to the genus extinction. This study arises the certainty that the evolution history of the genus is more complex than previously thought and that more studies are necessary to elucidate it, including a complete revision of the type material of Melissiodon dominans and Melissiodon schlosseri in the light of current knowledge of the genus, which will help to elucidate the attribution of the populations from Mokrá-Quarry. For the time being, the assemblage presented here is referred as Melissiodon aff. schlosseri.

Introduction

The extinct genus Melissiodon originally described by Schaub, 1920 is a rare fossil cricetid that existed from the Early Oligocene (MP23) to the Early Miocene of Europe (MN4). However, more recent research on the biostratigraphy in the Vallès-Penedès Basin (Catalonia, Spain) date the Miocene site of Sant Mamet, from which Melissiodon remains were recovered (Jovells-Vaqué & Casanovas-Vilar, 2018), to the beginning of the MN5 (Jovells-Vaqué, 2020; Jovells-Vaqué & Casanovas-Vilar, 2021). The genus name means “honeycomb tooth”, in reference to its peculiar molar morphology, which has slender crests enclosing numerous pits. The remains are very scarce and mostly fragmentary in the few European localities this genus has been recovered. It is further reported that the only record outside Europe is that of Melissiodon sp. from the Kargi-2 Anatolian assemblage at latest Oligocene-earliest Miocene (De Bruijn, Ünay & Hordijk, 2013). However, it comprised only a complete M3 and a fragmented m1. Melissiodon includes currently nine species (Hrubesch, 1957; Mödden, 1999), six of them recognized in Europe during the Oligocene (Schaub, 1920; Schaub, 1925; Freudenberg, 1941; Hrubesch, 1957). The most extensive work on the genus of this epoch was performed by Hrubesch (1957). The author reviewed most of the genus available material and erected three new taxa: two new species and one subspecies. However, Melissiodon was much more diverse during the Oligocene than in the Early Miocene, including four different lineages (see Hrubesch, 1957).

Regarding the Miocene only two species have been identified. Melissiodon schlosseri, the type species of the genus, was established long ago by Schaub (1925). It was based on one m1 and one m3 from Haslach (Germany, MN2). Hrubesch (1957) later suggested that an m3 from La Chaux in the MN2 Swiss Molasse Basin could belong to the same species. Werner (1994) and Engesser & Mödden (1997) then identified a very small assemblages of M. aff. schlosseri from the MN2 localities of Ulm-Uniklinik in Germany and La Chaux 7 in Switzerland, but a faunal list was only published from the latter site.

The second species is Melissiodon dominans, and was erected by Dehm (1950) in a rich collection from the MN3 Wintershof-West fissure-filling in Germany. This exceptional assemblage constitutes one of the largest collections recovered of the taxa, and it includes 40 M1, 36 M2, 8 M3, 36 m1, 39 m2 and 30 m3 dentary. It was then studied in detail by Hrubesch (1957), who assessed the intraspecific variation of the species from this locality, and the other Wintershof-Ost, Wütherich, and Schnaitheim German MN3 sites. These were not as rich as the type locality. In the slightly younger locality of Schnaitheim, significantly larger remains than the type population were found and referred to M. dominans. Remains of this species have also been found in other European MN3 localities, including: Austria (Mein, 1989); Switzerland (Bolliger, 1992); Spain (Crusafont, Villalta & Truyols, 1955; Agustí, 1981; Sesé, 1987; Jovells-Vaqué & Casanovas-Vilar, 2018); and France (Aguilar et al., 2003; Bulot, Antoine & Duranthon, 2009). However, the only record of M. dominans in the Czech Republic is from Ahníkov I MN3 site, and it was published without detailed description (Fejfar, Dvořák & Kadlecová, 2003).

Regarding the MN4 sites, although Melissiodon dominans has been identified, very few have the same richness as the MN3 localities. Ziegler & Fahlbusch (1986) described Melissiodon material from the Upper Freshwater Molasse sites of Rembach and Fortshart in Germany. These were attributed to M. dominans despite differences in size and morphology. The authors also studied remains of the much richer sites of Erkertshofen 1 and Petersbuch 2, but again there was no accompanying depiction. Moreover, Mein & Freudenthal (1981) and Bulot, Antoine & Duranthon (2009) described scarce remains of Melissiodon from the Vieux-Collonges in Rhône and Beón 2 at Montreal du Gers sites in France. Scant elements referred to M. dominans were located at the Montalvos 2 (Hordijk et al., 2015), Barranc de Campisano and Mas d’Antolino (Crespo, 2017), and Sant Mamet Spanish sites (Jovells-Vaqué & Casanovas-Vilar, 2018). Fejfar (1990) published measurements and drawings of some upper and lower first molars from the Early Miocene Czech localities of Dolnice 1, 2, 3 and Ořechov, but there was no information on remaining molars. Early Miocene MN4 finds located at the Mokrá-Quarry karstic site near Brno in the Moravian part of the Czech Republic (Fig. 1) constitute the latest fossil site from which Melissiodon remains have been recovered. Ivanov, Musil & Brzobohatý (2006) contains a detailed geological setting on Mokrá-Quarry and Bonilla-Salomón et al. (2021a) discuss its age based on the small mammal association. The genus Melissiodon has been identified in MWQ 1/2001 and MWQ 2/2003 fissures from the Western Quarry and MCQ 3/2005 from the Central one. Bonilla-Salomón et al. (2021a) described a small association of Melissiodon dominans from MWQ 1/2001 at the species level. However, M. dominans has become a “catch-all” species, because most of the remains found in Europe in MN3 and MN4 localities are referred to this species, despite its dubious attribution from the few remains recovered in most of the assemblages. The morphological characteristics of the species vary significantly in different localities. This is exemplified in the German Forsthart and Rembach assemblages differing from the type population. These also differ in the slightly younger localities, where M. dominans remains are notably smaller than the type population at most MN4 sites (a complete discussion is contained in the “remarks” section). Overall, several authors have identified populations which can be referred to different Melissiodon forms, but the scarcity of the remains prevented them to erect a new species (Mein & Freudenthal, 1981; Bulot, Antoine & Duranthon, 2009).

Figure 1 Location of Mokra-Quarry in Moravia, Czech Republic.

(A) Geographical position of Mokrá-Quarry. (B) Position of Mokrá Western and Central Quarries, as well as the location of MWQ 1/2001, MWQ 2/2003, MCQ 3/2005, MWQ 4/2018 sites. Source: Mapy.cz (modified); ©Seznam.cz, a.s., under license (CC-BY-SA 4.0).

The assemblage from Mokrá-Quarry is exceptional, since it constitutes one of the best documented findings of the genus in MN4 localities. This work describes all Melissiodon aff. schlosseri material available from the two MWQ 2/2003 and MCQ 3/2005 karst fissures from Mokrá-Quarry. This is followed by an assessment of morphological differences in MN3 and MN4 European assemblages, discussion of the evolution of the genus during the Early Miocene and proposed hypotheses.

Material & Methods

The micromammal fauna from MWQ 2/2003 and MCQ 3/2005 was discovered during field campaigns led by M. Ivanov and R. Musil from the Masaryk University in Brno (Czech Republic) during 2002–2005. Fossil remains mixed in sand and clays were obtained by washing in 0.5 mm mesh sieves (Ivanov, Musil & Brzobohaý, 2006). The field campaigns were approved by the Masaryk University under project number: MUNI/31 8016 “Systematic survey of sediments of karst fissures in the area of the Mokrá Quarry”. The material is currently housed under SMM/009-09-11/ 372009, Pal. 3000–3910 inventory numbers in the Department of Geological Sciences collections at the Masaryk University Faculty of Science. The upper cheek teeth are indicated as M1, M2, M3 and the lower cheek as m1, m2 and m3. Dental terminology follows Jovells-Vaqué & Casanovas-Vilar (2018), with some modifications (Fig. 2). Additional crests and cristids are named based on their position relative to the main cusp or cuspid from which they run (e.g., anterior labial arm of entoconid).

Figure 2 Interpretative drawings and terminology of the upper and lower first molars of Melissiodon aff. schlosseri from Mokrá-Quarry. (A) M1 (Pal. 3380). (B) m1 (Pal. 3373).

Abbreviations: Alpl, anterolophule; Alplc; anterolophule cusp; Alpld, anterolophulid; Alpldc, anterolophulid cuspid; Ast, anterostyle; E Mtl; Extra Metalophule spur; Ecld, Ectolophid; Enl, Entoloph; Etcd, Entoconid; Hc, Hypocone; Hpc, Hypoconule; Hpcd, Hypoconid; La Ac; Labial Anterocone; La Atcd, Labial anteroconid; Li Ac, Lingual Anterocone; Li Atcd; Lingual anteroconid; Li Postl; Lingual Posteroloph; Msc, Mesocone; Mscd, Mesoconid; Msl, Mesoloph; Msld, Mesolophid; Msst, Mesostyle; Mtc, Metacone; Mtcd, Metaconid; Mtl, Metalophule; Pac, Paracone; Postl, Posteroloph; Postld, Posterolophid; Posts, Posterosinus; Postsd, Posterosinusid; Prc, Protocone; Prl, Protolophule; Prs, Protosinus; Prst; Protostyle; Prtsnsd, Protosinusid; Ptcd, Protoconid; Ptcd h arm, Protoconid hind arm; S, Sinus; Sd, Sinusid. Modified from Jovells-Vaqué & Casanovas-Vilar (2018).

Measurements were taken with a Carl Zeiss Stemi 305 microscope and the Carl Zeiss W-PI 10x/23 Microscope Focusable Eyepiece. These provided the maximum length and width (L × W) of the occlusal surface of cheek teeth in mm. Micrographs were taken with a Quanta FEG 250 Scanning Electron Microscope (SEM) at the Institute of Electrical Engineering of the Slovak Academy of Sciences (SAS) at Bratislava (Slovakia). All teeth are figured as left elements.

The following standards are adhered to: the Miocene time scale and biostratigraphy are based on the International Chronostratigraphic Chart (Cohen et al., 2020), the Central and Eastern Paratethys boundaries follow Gozhyk et al. (2015) and Kováč et al. (2018); MN-zones for Western (left) and Central (right) Europe are as in Steininger (1999) and Hilgen, Lourens & Van Dam (2012). The age and chronological position of the localities in the text and figures are based on all the following: Bolliger (1992); Steininger (1999); Bulot & Ginsburg (1996); Aguilar et al. (2003); Bulot, Antoine & Duranthon (2009); Reichenbacher et al. (2013); Ruiz-Sánchez et al. (2013); Hordijk et al. (2015); Prieto et al. (2018); Prieto et al. (2022) and Jovells-Vaqué & Casanovas-Vilar (2021).

Systematic Palaeontology

Order: RODENTIA Bowdich, 1821	
Family: CRICETIDAE Fischer [von Waldheim], 1817	
Subfamily: CRICETOPINAE Matthew & Granger, 1923	
Tribe: MELISSIODONTINI Schaub, 1925	
Genus: MelissiodonSchaub, 1920	
Melissiodon aff. schlosseriSchaub, 1925	

Remarks on the systematics

The formal definition of Melissiodontidae at the family level was published by Schaub (1925), after the author earlier indicated that the clade deserved a family rank (Schaub, 1920). However, many studies hesitated to include the genus Melissiodon in the family Cricetidae (e.g., Freudenthal, Lacomba & Malagón, 1992; Kristkoiz, 1992) because of its aberrant skull features and the derived morphology of the cheek teeth. Therefore, this genus was the only member of the subfamily Melissiodontinae for most of the last century. Later, Ünay-Bayraktar (1989) erected the genus Edirnella based on a few isolated upper cheek teeth from Kocayarma MP25 Middle Oligocene locality in the Turkish Thrace Basin. This genus was mainly accommodated into the subfamily Melissiodontinae because it shares a cusp between the anterocone complex and the protocone with Melissiodon. This cusp is referred to as the Melissiodon cusp by Ünay-Bayraktar (1989) and Wessels et al. (2018) and as the protostyle by Jovells-Vaqué & Casanovas-Vilar (2018). Ünay-Bayraktar (1989) also grouped the Melissiodontinae subfamily containing Melissiodon and Edirnella and the Paracricetodontinae with its Paracricetodon and Trakymys into the family Melissiodontidae. This was strongly criticized and avoided in subsequent works by Freudenthal, Lacomba & Malagón (1992); Kristkoiz (1992) and Kalthoff (2006). New discovered remains from Süngülü Late Eocene deposits in the Lesser Caucasus of Turkey were referred to Edirnella and included in the Melissiodontinae subfamily (De Bruijn et al., 2003). Mogilia represents the third genus belonging in the subfamily Melissiodontinae, which contains the two M. miloshi and M. lautus species of the Eocene-Oligocene of Serbia (Wessels et al., 2018).

The Cricetopsinae subfamily was originally established by De Bruijn & von Koenigswald (1994), but McKenna & Bell (1997) amended it to Cricetopinae, and this is followed in our work. The phylogenetic analyses performed by Maridet & Ni (2013) recovered Melissiodon into Cricetopinae subfamily, being Melissiodon the sister species of Selenomys, and the two would constitute the sister group of Mirrabella. The monophyletic group including three genera was called Melissiodon clade. Maridet & Ni (2013) included Cricetops, Meteamys, and Selenomys genera into Cricetopinae. However, the most remarkable reallocation was that Edirnella was recovered into subfamily Pseudocricetodontinae, and consequently, more closely related to the genera Pseudocricetodon, Adelomyarion and Raricricetodon. This proposal was not followed by Wessels et al. (2018), as they placed all the three genera, i.e., Edirnella, Melissiodon and Mogilia, into subfamily Melissiodontinae mainly based on the enamel microstructure, a key character omitted by Maridet & Ni (2013) in their phylogenetic study. Despite the debate regarding its phylogenetic position, Melissiodontinae represent, together with Atavocricetodon and Pseudocricetodon, one of the first muroids that have been recorded outside of Asia, and yet, these are still poorly known.

Measurements: see Table 1

(Figs. 3A–3AB)

Table 1 Teeth measurements (in mm) of Melissiodon aff. schlosseri from MWQ 2/2003 and MCQ 3/2005.

		MWQ 2/2003	MCQ 3/2005	
		N	Min.	Max.	mean	N	Min.	Max.	mean	
M1	L	8	3.11	3.28	3.20	1		3.39		
	W	9	2.06	2.24	2.14			2.17		
M2	L	6	2.08	2.34	2.22					
	W	6	1.76	1.99	1.84					
M3	L	1		1.55						
	W	1		1.79						
m1	L	3	2.39	2.52	2.46					
	W	3	1.78	1.98	1.88					
m2	L	3	2.20	2.31	2.31	1		2,25		
	W	3	1.72	1.83	1.78	1		1.64		
m3	L	4	2.35	2.48	2.42					
	W	5	1.58	1.72	1.69					

Figure 3 Scanning electron micrographs of Melissiodon aff. schlosseri from MWQ 2/2003 and MCQ 3/2005 in occlusal view.

(A) Fragment of left mandible with M1-M2 (Pal. 3381); (B) fragment of right mandible with M1-M2 (Pal. 3382); (C) fragment of left mandible with M1 (Pal. 3490); (D) fragment of right mandible with M1 (Pal. 3488); (E) left M1 (Pal. 3371); (F) right M1 (Pal. 3380); (G) fragment of right M1 (Pal. 3489); (H) left M1 (Pal. 3491); (I) left M1 (Pal. 3492); (J) left M1 (Pal. 3493); (K) right M1 (Pal. 3931); (L) right M2 (Pal. 3372); (M) right M2 (Pal. 3383); (N) right M2 (Pal. 3495); (O) left M2 (Pal. 3496); (P) left M3 (Pal. 3377); (Q) left m1 (Pal. 3373); (R) right m1 (Pal. 3374); (S) right m1 (Pal. 3494); (T) fragment of left m1 (Pal. 3903); (U) left m2 (Pal. 3375); (V) right m2 (Pal. 3497); (W) right m2 (Pal. 3498); (X) left m2 (Pal. 3930); (Y) fragment of right m3 (Pal. 3499); (Z) right m3 (Pal. 3376); (AA) right m3 (Pal. 3501); (AB) left m3 (Pal. 3502).

Studied material:

MWQ 2/2003: fragment of left maxilla with M1-M2 (Pal. 3381); fragment of right maxilla with M1-M2 (Pal. 3382); fragment of left maxilla with M1 (Pal. 3490); fragment of right maxilla with M1 (Pal. 3488); four left M1 (Pal. 3371, Pal. 3491, Pal. 3492, Pal. 3493); two right M1 (Pal. 3380, Pal. 3489); one left M2 (Pal. 3496); four right M2 (Pal. 3372, Pal. 3383, Pal. 3495); one left M3 (Pal. 3377); one left m1 (Pal. 3373); two right m1 (Pal. 3374, Pal. 3494); one left m2 (Pal. 3375); two right m2 (Pal. 3497, Pal. 3498); two left m3 (Pal. 3501, Pal. 3502); three right m3 (Pal. 3376, Pal. 3499, Pal. 3500).

MCQ 3/2005: one right M1 (Pal. 3931); one fragment left m1 (Pal. 3903); one left m2 (Pal. 3930).

Description

M1 (Figs. 3A–3K): The teeth have five cylindrical roots, with the two larger ones below the anterocone complex and the protocone. Two slimmer roots are below the metacone and the hypocone and the fifth root is conspicuously thinner and situated below the paracone. The anterocone complex is massive and protrudes anteriorly. It presents two cusps, the largest one placed in a centro-lingual position. Both cusps are separated by a deep and narrow notch, which is almost enclosed anteriorly in Pal. 3492 (Fig. 3I). The lingual anterocone is connected to the anterolophule. This cusp also has a lingual spur that merges with a distolabial crest running from the protostyle. Furthermore, it connects to the anterolophule, dividing the protosinus in 8 out of 11 teeth (Pal. 3491, Fig. 3H); in Pal 3492 (Fig. 3I), this lingual spur merges with the anterolophule before the anterolophule cusp, leaving the protostyle isolated and the protosinus complete; in Pal. 3489 (Fig. 3G), it ends before reaching the spur from the protostyle, and in Pal. 3382 (Fig. 3B), it reaches the anterolophule cusp without connecting to the protostyle. In addition, Pal. 3493 (Fig. 3J) has a large spur from the protostyle that reaches the labial spur of the lingual anterocone. The anterior spur of the protostyle connects to a spur running lingually from the lingual anterocone enclosing a small sinus in 7/11 teeth (Pal. 3371, Fig. 3E). This crest is specially developed in Pal. 3490 (Fig. 3C) and the rest of the specimens show a different degree of development of the spurs without fully enclosing the small sinus (i.e., Pal. 3491, Fig. 3H). The labial anterocone is also large: it connects to the lingual anterocone through a lingual spur and to the two anterior paracone spurs, enclosing the anterosinus. The anterior lingual paracone spur has a short crest running labially in Pal. 3380 (Fig. 3F). In Pal. 3489, 3491 and 3493 (Figs. 3G, 3H, 3J) there is a short crest running lingually towards a short labial spur from anterolophule but without connecting to it. A small anterostyle is present in Pal. 3381, 3382, 3371 and 3493 (Figs. 3A, 3B, 3E, 3J); it is very strong and isolated from the other cusps in Pal. 3492 (Fig. 3I). Pal. 3931 has a short transverse crest between the paracone and the labial anterocone (Fig. 3H). The protolophule is defined by a labial spur of the protocone, which is slightly longer than the lingual one of the paracone. The mesocone is well-developed and connected to the protolophule through a short anterior arm (10/11) or isolated from it (1/11; Pal. 3490, Fig. 3C). The posterior arm of the mesocone runs labially and connects to the entoloph, except for Pal. 3490 (Fig. 3C), in which seems to be isolated.

Although the morphology of the mesoloph is also variable, it always runs obliquely to the labial end of the tooth. It connects to the posterior arm of the mesocone, and in three specimens, it is long enough to reach the anterior arm of the mesocone (Figs. 3E, 3F, 3G). The entoloph runs labially and joins the metalophule. The mesoloph area is highly variable. In Pal. 3382 and 3488 there is a single connection to the mesoloph, but closer to the mesocone (Figs. 3B, 3D). In Pal. 3488, the extra metalophule spur runs distolabially and connects to the metacone (Fig. 3D); however, this crest is much shorter in Pal. 3491 and Pal. 3931 (Figs. 3H, 3K) in anteroposterior direction. The remaining teeth (7/11) have an extra metalophule spur that is joined to the mesoloph, enclosing a small pit that can be narrow (Pal. 3381, Fig. 3A) or wide (Pal. 3371, Fig. 3E). Pal. 3382 (Fig. 3B) is the only tooth that does not show an extra metalophule spur. Both the paracone and the metacone show labial spurs that connect with the mesoloph. This connection protrudes labially and is specially marked in Pal. 3382 (Fig. 3B). The sinus is partially open. The posteroloph is low and connects to the posterior spur of the metacone. In 7/11 teeth it runs labially surpassing the posterior spur of the metacone (Fig. 3F). In 9/11 teeth a hypoconule is present (Fig. 3F). All specimens show a short lingual posteroloph from the hypocone.

M2 (Figs. 3L–3O): All molars have a rectangular outline. The anterolabial is the thinnest of its four roots. The lingual anteroloph is short but very robust and anterocone-like in 6/7 teeth, and very thin in Pal 3383 (Fig. 3M). The anterolophule reaches the anterior lingual spur of the paracone. There is a distinct anterolophule cusp in half the molars. Additionally, Pal. 3381 shows a small bump where the anterolophule meets the anterior lingual spur of the paracone (Fig. 3A). The labial anteroloph is connected to the labial anterior spur of the paracone in 6/7 teeth, but Pal. 3381 shows an incomplete labial anteroloph, leaving the anterosinus open (Fig. 3A). The protolophule is straight. The posterior lingual spur of the paracone is long in all teeth except for Pal. 3381 (Fig. 3A). It reaches the protolophule (3/6; Fig. 3L) or ends immediately before it (3/6; Fig. 3N). The protolophule merges with the mesoloph in Pal. 3381, 3382, and 3495 (Figs. 3A, 3B, 3N). The mesoloph and mesocone show high variability within the assemblage of Mokrá-Quarry. The mesocone is round. The anterior arm of the mesocone turns anteriorly and connects to the protolophule in two specimens (Figs. 3L, 3O). It runs labially and connects to the mesoloph in Pal. 3383 (Fig. 3M). However, in Pal. 3381 and 3495 the anterior arm of the mesocone runs labially and splits in two, connecting to the protolophule and to the mesoloph (Figs. 3A, 3N). A further deviation is the reduction of the anterior arm of the mesocone to a short spur that does not reach the mesoloph or the protolophule. This, however, is apparent only in Pal. 3382 (Fig. 3B). The posterior arm of the mesocone connects to the entoloph in all specimens except for Pal. 3372 where it is connected to the metalophule (Fig. 3L). Pal. 3495-3496 (Figs. 3N, 3O) highlight the extra spur from the posterior arm of the mesocone which reaches the mesoloph. This extra spur has a double connection to the entoloph and does not fully connect to the mesoloph in Pal. 3381 (Fig. 3A). The metalophule is always connected to the mesoloph. A mesostyle is absent and the ectoloph connects to the metalophule (Fig. 3L). The sinus is open in all teeth and the posteroloph reaches the posterior spur of the metacone. All teeth show a lingual posteroloph running from the hypocone (Fig. 3M).

M3 (Fig. 3P): This molar has two labial roots and a thicker lingual root. It is narrow and has a round outline. The labial anteroloph is thin and does not reach the paracone labial (Fig. 3P). There is a preserved very low lingual anteroloph that does not reach the base of the protocone, and both protocone and paracone are high and well-developed. The anterolophule is straight and reaches the anterior lingual spur of the paracone. The protolophule is fused with the metalophule, which runs from a very small metacone. The lingual paracone spur does not reach the protolophule. The entoloph is somewhat short and therefore does not reach the metalophule. The posterior arm of the hypocone connects with the metacone, and all the sinuses are open.

m1 (Figs. 3Q–3T): Both lingual and labial anteroconids are well-developed and separated by a deep anterior groove which is clearly visible in Pal. 3373 (Fig. 3Q). Despite its fragmentary preservation, Pal. 3494 has a short cristid running lingually from the labial anteroconid towards the lingual one, without fully attaching to it (Fig. 3S). The lingual anteroconid is massive and located slightly posterior to the labial anteroconid. It is connected to the metaconid through a short posterior spur. The metaconid lacks a well-developed anterolingual cristid, which leaves the anterosinusid open. The anterolophulid connects to both anteroconids and to the protoconid in all teeth. The anterolophulid connects to the labial anteroconid posteriorly. The anterolophulid cuspid is large and there is a short spur towards the metaconid, but without connecting to it. The labial anteroconid is large and bulge-like, and protrudes anteriorly. It is isolated from the protoconid as the protosinusid is open. The protoconid is large and its hind arm connects to the labial posterior spur from the metaconid, and reaches the lingual edge of the tooth (Fig. 3R). The anterior arm of the mesoconid connects to the protoconid hind arm in all specimens, showing a small bulge in Pal. 3374 (Fig. 3R). The mesolophid always starts at the anterior arm of the mesoconid, showing different degrees of development: in Pal. 3373 it is divided and does not reach the anterior arm of the entoconid (Fig. 3Q); in Pal. 3374 it turns slightly anterior, connecting to the protoconid hind arm and enclosing a small pit (Fig. 3R); in Pal. 3494 (Fig. 3S) it runs parallel to the protoconid hind arm towards the labial anterior spur of entoconid; in Pal 3903, the anterior arm of the mesoconid appears to run labially and the mesolophid is reduced to a short spur that ends next to the anterior arm of the entoconid without attaching to it (Fig. 3T). The ectolophid is short and attached to the mesoconid. The entoconid has the labial cristids much better developed than the lingual ones, which are almost absent. The labial posterior cristid reaches the posterolophid. There is a well-developed posteriorly oriented spur running from the posterolophid (Fig. 3S). The sinusid is open labially. A well-developed ectomesolophid that almost reaches the mesoconid is present in Pal 3494 (Fig. 3S). However, in Pal 3903 there is a small bump instead (Fig. 3T).

m2 (Figs. 3U–3X): Pal. 3498 is damaged anteriorly and lingually (Fig. 3W). The four teeth preserve a very short labial anterolophulid. The protosinusid is open, and the lingual anterolophid is well-developed and connects to an anterior labial spur running from the metaconid. There is no anterior lingual cristid from the metaconid, and this leaves the anterosinusid open. The protoconid hind arm is long and runs distolingually reaching the labial edge. The posterior spurs of the metaconid and the anterior spur from the entoconid attach to the protoconid hind arm. The mesoconid is well-developed and its short anterior arm runs anterolabially to connect to the protoconid hind arm. The mesolophid preserves different configurations: in Pal. 3375, a very short spurs starts from the protoconid hind arm but does not reach the mesoconid (Fig. 3U); in Pal. 3497, a short mesolophid starts from the centre of the mesoconid and merges with the protoconid hind arm (Fig. 3V); in Pal. 3498, a long mesolophid, starting from the centre of the mesoconid runs parallel to the protoconid hind arm and connects to the anterior labial spur of the entoconid (Fig. 3W); in Pal. 3930 (Fig. 3X), the short mesolophid merges with the protoconid hind arm as seen in Pal. 3497. The ectolophid is short and preserves a short spur in 3375 (Fig. 3U), but it is long in Pal. 3390 (Fig. 3X). The posterolophid is very low and connects to the two posterior spurs of the entoconid. The sinusid is open.

m3 (Figs. 3Y–3AB): The molars are elongated with a rounded posterior side. There are two roots, with the posterior root much broader than the anterior. The protoconid connects to an anterior labial spur which runs from the metaconid through a well-developed anterolophulid that runs straight. Although some teeth, such as Pal. 3376 and 3502, have a thickening on the anterolophulid there is no distinct anteroconid (Figs. 3Z, 3AB). The protosinus is open, and the protoconid hind arm reaches the labial edge and connects to the lingual posterior spur of the metaconid and the anterior one from the entoconid. In addition, Pal. 3376 and 3501 also have a posterior labial spur of the metaconid (Figs. 3Z, 3AA). The most variability occurs in the mesoconid, as in m2. While Pal. 3376 and 3502 present a small but distinct mesoconid (Figs. 3Z, 3AB), the remainder have no distinguishable bulge. The hypolophulid is short and directed distolingually (Fig. 3Y). Pal. 3501 has a hypolophulid divided into two very short spurs (Fig. 3AA). The posterolophid becomes progressively lower and attaches to the base of the entoconid. There is no posterior spur running from this cuspid. The sinusid is open.

Discussion

The Mokrá-Quarry Melissiodon remains described herein differ from the Melissiodon dominans at all MN3 and MN4 localities. For instance, it only approaches the minimum MN3 M. dominans size-range, and this includes the Wintershof-West type population (Fig. 4). The Mokrá-Quarry population is also smaller than those at most MN4 sites, except at Rembach and Forsthart German sites. There are also marked morphological differences. Hrubesch (1957) noted Melissiodon’s wide range of morphological variability, especially in the first upper and lower molars, where the stronger morphological differences occur. However, our study reveals clear differences in second and third molars, in addition to those reported.

Figure 4 Length/Width scatter plot (in mm) of the upper and lower molars of different Melissiodon taxa from various localities during the Early Miocene.

Measurements of Melissiodon taxa were obtained from the following publications: Hrubesch, 1957; Mein & Freudenthal, 1981; Ziegler & Falhbusch, 1986; Fejfar, 1990; Werner, 1994; Hordijk et al., 2015; Jovells-Vaqué & Casanovas-Vilar, 2018.

The Mokrá-Quarry Melissiodon assemblage resembles the single m1 and m3 Haslach M. schlosseri remains, and the M. aff. schlosseri mandible with m1-m3 from Ulm-Uniklinik (both from Germany, MN2). This resemblance is the large cuspid-like labial anteroconid that protrudes anteriorly in the m1. However, while the connection of this cuspid to the anterolophid is posterior in the Mokrá-Quarry population, it is more lingual in the German Melissiodon sample (Schaub, 1925: plate 4, fig. 16; and Werner, 1994: fig. 27a). However, the under sampling of M. schlosseri and M. aff. schlosseri populations prevents determination whether it is morphological difference or intraspecific variability. In addition, the overall robust m1 cuspid pattern in the anteroconids, main cuspids and mesoconid is a shared structure in the Mokrá Melissiodon and M. schlosseri and M. aff. schlosseri populations. While the single recovered M. aff. schlosseri m2 resembles the overall Mokrá-Quarry samples’ morphology, the posterior labial cristid of the metaconid accentuates differences. Although this always connects to the protoconid hind arm in the Mokrá teeth, the connection in the Ulm-Uniklinik m2 is to the the lingual posterior cristid of the metaconid (Werner, 1994: fig. 27b). In contrast, there is no marked difference in the m3’s.

The M. dominans type population from Wintershof-West presents a small labial anteroconid that is connected lingually to the anterolophulid and labially to the protoconid (Hrubesch, 1957), and this differs to the Mokrá-Quarry m1s’ anterior morphology. The Mokrá-Quarry m1 has a clearly developed cuspid-like labial anteroconid which protrudes anteriorly and is only connected to the anterolophulid by a posterior spur (Fig. 3Q; see also Bonilla-Salomón et al., 2021a: fig. 3A).

The Wintershof-West Melissiodon dominans m2 is somewhat larger. When the mesolophid is present here, it develops from the anterior arm of the mesoconid in most of the population, but it always begins from the mesoconid in Melissiodon from Mokrá-Quarry. The Melissiodon dominans m3 from the type locality is also larger than those from Mokrá-Quarry, while two of the latter have a posterior labial spur of the metaconid. This characteristic is absent in all Early Miocene M. dominans populations.

The Mokrá-Quarry Melissiodon M1 anterocone complex protrudes anteriorly and is massive compared with the M. dominans type population and other MN3 sites. The furrow between labial and lingual anterocones is narrower than any population of M. dominans. In addition, the lingual anterocone has a second posterior spur, which runs lingually and connects with a posterior spur of the protostyle and an anterior one from the anteroloph, thus dividing the protosinus. This feature, present in the vast majority of Mokrá-Quarry M1 is only rarely present in the type population and in single specimens from other MN3 sites: Ramblar 7 (Sesé, 1987), and Turó de les Forques 1 (Jovells-Vaqué & Casanovas-Vilar, 2018).

The recovered Mokrá-Quarry M2 is also clearly different to M. dominans from the type locality. It is narrower than the type population (Fig. 4), and the Mokrá-Quarry Melissiodon assemblage has a long posterior lingual crest from the paracone, which either connects to the protolophule or ends immediately before it. In contrast, the Wintershof-West M. dominans shows a very short posterior lingual spur of the paracone that never reaches the protolophule (Hrubesch, 1957: plate 3, figs. 8–11). The M3 is smaller than the type locality (Fig. 4) and the protolophule turns posteriorly merging with the metalophule towards a very small metacone. Therefore, it does not reach the paracone.

The Melissiodon dominans from Forsthart and Rembach (Germany, MN4) preserve an isolated, cristid-like, labial anteroconid (see Ziegler & Fahlbusch, 1986: plate 10, figs. 4–6). This anterior m1 morphology differs from the M. dominans from the type locality and even more with the robust cuspid-like labial anteroconid in the Mokrá-Quarry Melissiodon populations. Unfortunately, Ziegler & Fahlbusch (1986) did not consider this morphological character different enough to erect a new species. Although the Mokrá-Quarry, Forsthart and Rembach sites yielded similar small mammal assemblages (i.e., Democricetodon and Megacricetodon), the displayed Melissiodon morphology of this character in these sites is completely different. This suggests that the more robust labial and lingual anteroconids identifies different evolutionary lines in the genus. Moreover, the presence of well-developed anteroconids in the older Melissiodon populations (i.e., Haslach and Ulm-Uniklinik) could indicate the development of this trend in different lineages in different periods.

Although the Mokrá-Quarry Melissiodon m2 has a connection of the posterior labial cristid of the metaconid to the protoconid hind arm, this cristid never reaches the protoconid in arm in the Forsthart and Rembach populations. In addition, the mesolophid begins at the mesoconid, while it always begins at the anterior arm of the mesoconid in M dominans from these and other MN4 localities.

Ziegler & Fahlbusch (1986) considered that the presence of a labial spur of the mesoconid in all m3s is a derived character of Melissiodon dominans from MN4 sites. However, this feature is absent in all Mokrá-Quarry m3s, and its absence would indicate the presence of a different form of Melissiodon because the assemblage has a very similar age to those of the German sites.

In addition, the German Forsthart and Rembach M. dominans M1 appear to have a much more developed labial anterocone, and this is positioned more labially than in the Mokrá-Quarry’s Melissiodon assemblage. The German populations’ M1 also have no second connection between the lingual antercone and the anterolophule, and their M2 have a shorter posterior lingual spur of the paracone and no evident connection between the protolophule and the mesocone. This feature, however, is preserved in 5/7 teeth in Mokrá-Quarry (Figs. 3L–3O). The German assemblages’ M3s show the same differences to Mokrá-Quarry Melissiodon as the discussed type population.

Compared to Western European Melissiodon assemblages, a well-developed labial cuspid-like anteroconid in the Mokrá-Quarry population resembles the conditions of the single m1 reported from the MN5 site of Vieux-Collonges, (Mein & Freudenthal, 1981: plate 2, fig. 12), and the older and larger m1 from Sant Andreu de la Barca 1 (Jovells-Vaqué & Casanovas-Vilar, 2018; fig. 2c). It is noteworthy that the connection with the anterior spur of the anterolophulid is lingual in these sites, while it is posterior in the Mokrá-Quarry Melissiodon remains. However, a cristid-like labial anteroconid is clearly visible in the single m1 available from Montalvos 2 (Hordijk et al., 2015: fig. 2e), and this resembles the two German populations. In addition, recently published M. dominans material from Echzell also has a cristid-like labial anteroconid, but connected lingually to the anterolophulid (Jovells-Vaqué & Mörs, 2022).

The single M1 specimens recovered from Sant Mamet and Montalvos 2 are larger and do not have a second posterior spur from the lingual anterocone (Jovells-Vaqué & Casanovas-Vilar, 2018; Hordijk et al., 2015). Compared with other Czech sites (i.e., Dolnice 1, 2 ,3 and Ořechov), within the populations there are specimens that also possess a bulgy labial anteroconid (see Fejfar, 1990), and therefore resemble those from Mokrá-Quarry more than M. dominans from other MN3 and MN4 sites. In addition, most of the M1 assemblage depicted by Fejfar (1990) also show a second posterior spur of the lingual anterocone dividing the protosinus. However, that study only included the first upper and lower molars, and no information regarding the rest of the tooth row.

The single M3 and the m1 preserved portion (posterior side) from the Kargi 2 Turkish locality attributed to Melissiodon sp. differ greatly with the Mokrá-Quarry population. For instance, the M3 has a much more rounded outline and lacks the connection between protolophule and metalophule, which is present in Melissiodon from Mokrá-Quarry.

The Mokrá-Quarry Melissiodon remains have a greater overall resemblance to M. schlosseri than to M. dominans, especially in the m1’s anterior half. However, M. schlosseri was erected based on a single m1 and m3 from the MN2 Haslach locality and it is poorly understood. The M. dominans populations in MN3 and MN4 localities also have wide morphological variability, and it is quite difficult to establish the species’ differential characteristics. However, the material described here is referred to as Melissiodon aff. schlosseri. This is based on the above reasoning, and our current knowledge. In addition, the MWQ 1/2001 material ascribed to M. dominans by Bonilla-Salomón et al. (2021a) is here referred to as M. aff. schlosseri therein.

Evolutionary history of Melissiodon during the Early Miocene

The first phylogenetic hypothesis of the Melissiodon genus was proposed by Hrubesch (1957: fig. 125), in which he suggested the existence of various lineages during the Oligocene, while during the Early Miocene only two branches persisted. Although he also considered Melissiodon arambourgi a completely different lineage, the species has been proven synonymous with M. dominans (Agustí, 1981; Jovells-Vaqué & Casanovas-Vilar, 2018). Melissiodon dominans became the most abundant species during the Early Miocene MN3 and MN4 localities and reached the Burdigalian before its extinction during the latest Early Miocene. The second lineage present in the Early Miocene led to M. schlosseri. The recovered Haslach material was quite sparse, but Hrubesch (1957) considered it different to any other species of the genus. He based this on the specialization of the m1 anterior half, and especially on its stronger cuspids and well-developed bulge-like anteroconids. These differences were enough for him to suggest a different lineage with no connection to M. dominans. Since only Haslach’s discovery of these remains existed, this species was considered a completely isolated branch in European localities, with unknown origin and no continuation throughout the Early Miocene.

Werner (1994) later described a partial left mandible with m1-m3 from Ulm-Uniklinik (MN2), a slightly younger locality than Haslach, and referred it to Melissiodon aff. schlosseri. The author indicated a small difference in the labial anteroconid size. Although it was large, it was only somewhat smaller than the type species. Since the type species included only one m1, the smaller size could be explained by intraspecific variability.

Engesser & Mödden (1997) then reported two different Melissiodon taxa from La Chaux 7, which is an MN2 Switzerland site with similar age to Haslach. The remains were attributed to M. aff. schlosseri and M. cf. dominans, but no descriptions or measurements of either taxon were provided. La Chaux 7 is the only current site where two different taxa of the genus have been recorded, and no other remains of Melissiodon related to M. schlosseri or this lineage of the genus (sensu Hrubesch, 1957) have been discovered at MN3 or MN4 localities.

The Mokrá-Quarry Melissiodon aff. schlosseri has highly developed m1 lingual and labial anteroconids, and it has greater development than M. dominans from the MN3 type locality. It is also arranged differently than in M. dominans from other MN4 Central European localities. In addition, the poorly known Melissiodon schlosseri has m1 anteroconids that resemble the size and development of Mokrá-Quarry M. aff. schlosseri.

The overall differences between the Mokrá-Quarry assemblage and M. dominans (see Discussion section above) imply that this material could be a younger species derived from M. schlosseri. However, our M. schlosseri knowledge depends on a very few teeth, and we consider that type material revision is required before it is possible to recognize a new species.

Although the high diachrony between European regions prevents a more straightforward correlation (e.g., Steininger, 1999; Gozhyk et al., 2015), the following figure herein depicts the biochronologic distribution of Melissiodon taxa during the Early Miocene (Fig. 5). If the Mokrá-Quarry Melissiodon aff. schlosseri is closely related to M. schlosseri, this would imply that the lineage survived in MN4 localities. However, the evolution and distribution of this lineage at MN3 localities still remains unclear.

Figure 5 Biochoronologic distribution of Melissiodon remains across Europe during the Early Miocene.

Black squares represent localities in which Melissiodon remains have been found, while white squares denotes localities in which Melissiodon remains with a cuspid-like labial anteroconid have been found. In the Dolnice 1–3 and Ořechov, m1 with both cuspid-like and cristid-like labial anteroconid have been found. The positions of MN4 Czech localities are speculative, since no biostratigraphic data for all localities is available. Age and chronological position of the localities mentioned in the figure is based on works of Bolliger (1992); Steininger (1999); Bulot & Ginsburg (1996); Aguilar et al. (2003); Bulot, Antoine & Duranthon (2009); Reichenbacher et al. (2013); Ruiz-Sánchez et al. (2013); Hordijk et al. (2015); Prieto et al. (2018); Prieto et al. (2022); Jovells-Vaqué & Casanovas-Vilar (2021). Melissiodon data was obtained from the following works: Hrubesch (1957); Mein & Freudenthal (1981); Ziegler & Falhbusch, (1986); Werner (1994); Hordijk et al. (2015); Jovells-Vaqué & Casanovas-Vilar (2018); and IBS 2021.

The single m1 from the MN3 Sant Andreu de la Barca has a similar anteroconid developmental pattern (Jovells-Vaqué & Casanovas-Vilar, 2018: fig. 2c), despite being larger than M. aff. schlosseri and M. schlosseri (see Fig. 4). The species-level taxonomical assignment of this single molar in the evolution of the genus is still uncertain, but it could indicate the presence of more taxa at MN3 localities besides M. dominans.

The m1s in MN4 Melissiodon assemblages are also smaller than in the type population. In addition to being smaller, the Melissiodon population from the German sites have a completely different m1 anteroconid area arrangement (Ziegler & Falhbusch, 1986), as noted in the Discussion section. The size increase in the anteroconid area is a noted common trend in the M. dominans lineage (Hrubesch, 1957). That, together with the complete lack of a labial anteroconid could be indicative of a different taxon. In addition, our recent inspection of the material from Dolnice and Ořechov (IBS 2021, personal observation) attributed to M. dominans by Fejfar (1990) reveals two different m1 morphotypes in those MN4 populations. These were the Mokrá-Quarry Melissiodon aff. schlosseri cuspid-like labial anteroconid and the German sites’ M. dominans cristid-like labial anteroconid. There was also clear predominance of the latter morphotype.

It is noteworthy that the m1 morphotype in the Mokrá-Quarry and other Czech localities is not restricted to Central Europe or MN4 sites. Mein & Freudenthal (1981) recovered a single m1 from Vieux-Collonges (France; MN5) which clearly resembles the Mokrá-Quarry M. aff. schlosseri. The authors considered the morphological differences in Melissiodon assemblages to be sufficiently distinct. Despite the insuficient material prevented them from erecting a new species, the morphological variability of the genus during the latest Early Miocene, prior to its extinction appears to be higher than previously acknowledged.

Besides, the alpha-taxonomy of Melissiodon populations in MN4 sites is complex, and the lineage leading to Melissiodon dominans remains poorly known. The existence of a tendency towards size increase between the type locality (Wintershof-West) and younger MN3 sites has been confirmed by several publications: Ramblar, Bañon and Turó de les Forques 1 sites in Spain (Sesé, 1987; Jovells-Vaqué & Casanovas-Vilar, 2018) and Beaulieu 2B and Jauquet in France (Aguilar et al., 2003; Bulot, Antoine & Duranthon, 2009). However, M. dominans from MN4 assemblages are usually smaller than those of the type locality (Fig. 4), which has also been noticed by other authors (Ziegler & Falhbusch, 1986; Bulot & Ginsburg, 1996; Bulot, Antoine & Duranthon, 2009). Overall, there are only a few exceptions in MN4 localities where Melissiodon dominans is larger than in other coeval sites. One of them is Montalvos 2, where M. dominans is larger than those from most MN4 localities where Democricetodon and Megacricetodon are also found (Hordijk et al., 2015). While these Montalvos 2 remains accord with the Wintershof-West M. dominans size-range, their m1 morphotype is similar to those of the Forsthart and Rembach MN4 sites. In addition, while the Dolnice 1 site contains two m1s similar to Montalvos 2 size, and are clearly larger than the remaining assemblage, the m1s have a cuspid-like labial anteroconid rather than the cristid-like configuration found in Montalvos 2.

The above evidence confirms Melissiodon dominans’ complex evolution, and therefore, the coexistence of two different lineages in the latest Early Miocene cannot be discounted. These comprise one that slightly increases in size throughout the MN3 (Aguilar et al., 2003); and a smaller lineage that replaced the larger one in most Central and Western Europe MN4 localities. In any case, the assessment of whether these represent different Melissiodon species or different populations within M.dominans is beyond the scope of this work. The small M. dominans populations and the scarcity of European fossil localities in the Early Miocene have made Melissiodon dominans into a catch-all species. Therefore, the complete revision of all Miocene localities with Melissiodon remains should elucidate the relationships between the different forms and species that existed in the European Early Miocene.

Paleoecological implications

Historically, the presence of Melissiodon has been linked to wooden areas and somewhat humid conditions (Van der Weerd & Daams, 1978). According to Sesé (1987), Melissiodon enters the Iberian Peninsula during a change from a relatively dry to a relatively wet climate (zone Z; Daams & Van der Meulen, 1984). In addition, it is no longer recorded in the Calamocha area beyond zone A, when the climate is presumed to have regained aridity.

The Mokrá-Quarry sites are characterized by its karstic conditions, together with patches of forest and swampy areas (Ivanov, 2008; Ivanov, Musil & Brzobohatý, 2006; Ivanov et al., 2017; Ivanov et al., 2020; Sabol et al., 2007; Luján et al., 2017; Luján et al., 2021; Bonilla-Salomón et al., 2021a). This concurs with conditions that presumably favored Melissiodon presence. Jovells-Vaqué & Casanovas-Vilar (2018) add that rich Melissiodon collections from the Vallès-Penedès Basin come from localities with diverse and abundant forest-dwelling rodents, such as dormice and tree-squirrels. We concur with this because several genera of sciurids adapted to arboreal lifestyle have also been recovered in the Mokrá-Quarry (Bonilla-Salomón et al., 2021b).

The existence of different Melissiodon lineages in the Early Miocene could also imply preference for specific palaeoecological conditions. However, there is no known locality during the Early Miocene where two different species have certainly been proven. La Chaux 7 (Switzerland, MN2), provides the only record of Melissiodon aff. schlosseri and M cf. dominans (Engesser & Mödden, 1997). However, without detailed descriptions, these attributions remain dubious. The high degree of specialization in the first upper and lower molars and clear difference between morphotypes are most likely related to slightly different feeding behaviors and consequent preference for those specific habitats.

Mein & Freudenthal (1981) suggested frugivorous diet for Melissiodon and ventured that the genus may be arboreal. However, Hordijk et al. (2015) hypothesized that Melissiodon was a ground-dweller genus that fed on invertebrates (mainly earthworms). They based this assumption on the similarity of teeth morphology to extant shrew rats from Sulawesi and the Philippines. This assumption was also supported by Wessels et al. (2018), suggesting that subfamily Melissiodontinae (sensu Wessels et al., 2018) would feed on small invertebrates. However, the main obstacle to interpreting feeding behavior and paleoecological preferences is the paucity of Melissiodon localities.

Conclusions

Herein we present the new MWQ 2/2003 and MCQ 3/2005 Mokrá-Quarry Melissiodon remains attributed to M. aff. schlosseri. These constitute one of the best documented assemblages of this genus in MN4 localities known so far. This population differs from other known Melissiodon assemblages in MN3 and MN4 localities in their first lower molars which have strong cuspids and well-developed labial and lingual anteroconids. This unique morphology soundly recalls that of the poorly known M. schlosseri from MN2 localities.

In addition, Melissiodon dominans from some German MN4 assemblages have a very reduced labial anteroconid. This disrupts Melissiodon species’ evolutionary trend of increasing the anteroconid size throughout the late Oligocene and the Early Miocene. Therefore, its assignation to M. dominans is dubious, and requires revision. Moreover, the apparent trend in size increase in M. dominans from MN3 localities (with special relevance in the population from Schnaitheim) is abruptly interrupted in MN4 assemblages. Thus, the existence of different lineages in the Early Miocene cannot be discounted.

In conclusion, the apparent diversity of poorly recorded taxa that lived during the Miocene suggests a more complex evolutionary history than previously acknowledged. A complete revision of the genus is required to clarify its diversification and the species’ relationships in the Early Miocene. In addition the Melissiodon remains from the Czech sites of Ahníkov I and II, Dolnice 1-3 and Ořechov need detailed study to understand the genus diversity. Finally, the presence of several species in the Central European Early Miocene could indicate preferences for different ecological conditions.

Supplemental Information

Supplemental Information 1 Melissiodon Mokrá Raw Data

Click here for additional data file.

The authors are very grateful to I. Kostič for taking the SEM images of the dental remains. The authors also thank Raquel López-Antoñanzas, handling editor, as well as three anonymous reviewers for their constructive comments that greatly improved the article.

Additional Information and Declarations

Competing Interests

Author Contributions

Field Study Permissions

Data Availability

The authors declare there are no competing interests.

Isaac Bonilla-Salomón conceived and designed the experiments, performed the experiments, analyzed the data, prepared figures and/or tables, authored or reviewed drafts of the article, and approved the final draft.

Stanislav Čermák analyzed the data, authored or reviewed drafts of the article, and approved the final draft.

Àngel H. Luján analyzed the data, prepared figures and/or tables, authored or reviewed drafts of the article, and approved the final draft.

Sílvia Jovells-Vaqué analyzed the data, authored or reviewed drafts of the article, and approved the final draft.

Martin Ivanov analyzed the data, authored or reviewed drafts of the article, and approved the final draft.

Martin Sabol conceived and designed the experiments, analyzed the data, authored or reviewed drafts of the article, and approved the final draft.

The following information was supplied relating to field study approvals (i.e., approving body and any reference numbers):

Field experiments were approved by the Masaryk University under the project number: MUNI/31 8016.

The following information was supplied regarding data availability:

The raw data is available in the Supplemental File.

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
