# Peer review of "Early Miocene remains of Melissiodon from Mokrá-Quarry (Moravia, Czech Republic) shed light on the evolutionary history of the rare cricetid genus"

_PeerJ, doi:10.7717/peerj.13820_

## Round 0.1 · original submission · Major Revisions

Dear author,

Your manuscript entitled “Early Miocene remains of Melissiodon from Mokrá-Quarry (Moravia, Czech Republic) shed light on the evolutionary history of the rare cricetid genus” which you submitted to PeerJ, has been reviewed. The reviewers' comments are included at the bottom of this letter. After carefully reading your manuscript nd the reviewer’s comments, I think that, subject to major revisions, your paper will be suitable for publication. Please consider all suggestions below and make appropriate revisions to your paper. Please, pay particular attention in including a comparison with the different Miocene populations, in justifying the allocation of your specimens, in updating the age of localities as requested by reviewers 2 and 3 and in editing the English of the manuscript.

Once again, thank you for submitting your manuscript to PeerJ and I look forward to receiving your revision.

If I can be of any assistance please do not hesitate to contact me.

Best wishes,

Raquel López-Antoñanzas

Reviewer 1 ·

Basic reporting

The manuscript titled “Early Miocene remains of Melissiodon from Mokrá-Quarry (Moravia, Czech Republic) shed light on the evolutionary history of the rare cricetid genus” by Bonilla-Salomón et al. presents a new population Melissiodon from the early Miocene of Mokrá-Quarry in Czech Republic. The manuscript includes a detailed description of the fossil teeth (focussing on quantifying the morphological variability) and is supported by precise measurements (including scatter diagrams of compared sizes) and good quality illustrations (SEM pictures). Following the description, the authors discuss the evolutionary trends and the possible ecological features of the genus during the early Miocene. Finally, the manuscript is overall well organised and the cited references are up to date.
The study is interesting and improves our knowledge of this, indeed, poorly known genus; it can be publish with minor revisions (see suggestions below):

- Line 98: genera misspelled: Adelomyarion and Raricricetodon
- Line 140: “referred” rather than “ascribed”
- Line 151: italic missing for dominans
- Line 161: “exceptional” not “exceptionally”
- Line 170: “led” instead of “leaded”
- Line 171: delete “The”
- Line : 270: “is turned” or “turns”
- Line 280: “Distinct” not “Disctinct”; additionally what do you mean by “No distinct mesostyle is discerned” ? Is the mesostyle absent or does something prevent from seeing it ?
- From line 353 to 459: The discussion is long. I would suggest waiting until the end of the discussion to summarize the arguments and then naming the population M. aff. dominans. I would also explain at end of the discussion why not naming a new species and why not M. aff. schlosseri.
- Line 367: who are “the authors”? previous authors or the authors of the present manuscript?
- Line 369: Nevertheless
- Line 387: those
- Line 470: replace “is” by “being”
- From line 571 to 598: all italics are missing for genera and species names
- Line 581: not all sciurids are arboreal, please precise

Experimental design

no comment

Validity of the findings

no comment

Additional comments

no comment

Reviewer 2 ·

Basic reporting

This article describes new material from a very specialized murioidea species that is poorly known and therefore interesting for the scientific community.
The manuscript presents a classical structure. The language used throughout the article needs to be reviewed as it contains numerous mistakes that should be corrected and that, in some cases, make it difficult to understand the meaning of the sentences.
The framework of the study is, in general, well established, although a more detailed introduction of the evolutionary framework of the genus in periods prior to the Miocene might improve to understand better the main context. The references are adequate, although a citation, made in the text, is missing in the list of references. The figures are all relevant and of good quality and the data included are adequate.

Experimental design

The study has a classic design and addresses taxonomical questions in an adequate way.
The study, as indicated above, is relevant and presents new data from undescribed populations of Melissiodon, a rare genus from the Oligocene and Miocene of Eurasia. The description of the new material are very detailed but sometimes it is difficult to understand them since part of the terminology of the dental elements used in the study is not indicated in figure 2.

Validity of the findings

Regarding the structure of the discussion, it would be appreciated if, instead of comparing each of the morphological characters shown in Mokrá material with those seen on the other populations of the Miocene, a complete morphological comparison with the different Miocene populations were done. In this way it would be easier to understand what the differences between populations are and it would be easier to assess the similarity or difference between populations, being their taxonomic identity more evident.
Regarding the taxonomy of this work, I do not quite understand why the Mokrá material is described as M. aff. dominans, if throughout the text it is made explicit that it is clearly a form not related to this species. Strong resemblances to M. schlosseri are proposed, so if they think it could be this species it should be assigned to M. cf. schlosseri. If, on the contrary, they suspect that it may be a new species but do not want to establish it until a more in-depth study of that species is completed, they should name the population of Mokrá as M. aff. schlosseri. In any case, as I have indicated previously, the discussion should be restructured so that the similarities and differences with the type populations of these species are better established in order to better assess their affinities.

Additional comments

Figure 5 must be corrected since the ages of the localities and their correlations are erroneous in light of recent works (Van der Meulen et al 2011, Reichenbacher et al 2013, Prieto and Rummel 2016, etc...). If you do not agree with these correlations, you should discuss why and what this new proposal is based on.
In the attached file several corrections and extra comments have been proposed.

Annotated reviews are not available for download in order to protect the identity of reviewers who chose to remain anonymous.

Reviewer 3 ·

Basic reporting

The authors present a precise work and describe in detail an important fossil collection attributed to the rodent Melissiodon (Lower Miocene, Czech Republic).

Their work is divided into 5 parts:

1 The introduction briefly introduces the genus, but expands on the history of the supra-generic affiliation of Melissiodon. A second chapter, entitled "the genus Melissiodon", takes up the genus again, commenting on its geographical and stratigraphic distribution.
(see § GENERAL COMMENTS)

2 a short § Material & Methods. The information necessary for a good understanding of the article is included, but it needs implementations (see § GENERAL COMMENTS)

3 A systematic section with very detailed descriptions.

4 The Discussion includes in a first part a comparison of the morphometric characteristics of the populations assigned to Melissiodon in Europe. On the basis of these results a sub-section proposes different hypotheses concerning the evolution of the genus in the Miocene (see § GENERAL COMMENTS). They end with paleoecological notes.

5-The conclusion summarises the main results and opens the way for further research

I'm not in a good position to comment on the quality of the English, but I think some smoothing is necessary.

Similarly, minor errors appear throughout the text (spaces, Melissiodon not in italics), but these are easy to detect after re-reading.

The list of references needs to be checked (e.g. De bruijn et al. 2013 is missing).

The figures are of good quality (see § GENERAL COMMENTS).

Experimental design

The authors use a standard but very detailed method of morphometric comparison. The arguments of the authors are then, from a taxonomic point of view, well argued.

The authors refer to five references to support their chrono- and biostratigraphic work. I think that a more precise discussion on the age of important localities is necessary, especially when the work proposes to follow the evolution of the genus during the lower Miocene (see § GENERAL COMMENTS)

Validity of the findings

The strongness of the article is the detailed description of the fossils, which allows us to demonstrate a great (taxonomic?) variability within the European populations at the end of the Miocene. The data necessary for a good understanding of the article have been provided.

I do have one major criticism. This concerns the correlations of the localities located in the text and presented in figure 5. I have detailed in the § GENERAL COMMENTS

Additional comments

This genus Melissiodon disappears at the end of the Lower Miocene, although it survived the cricetid vacuum. The locality in the Czech Republic studied is of this age. From a paleogeographical point of view the locality is also important. In my opinion, it is very interesting to work on this genus, whose species M. dominans probably does not correspond to a biological reality, as the authors state. This work is therefore also important for the understanding of faunal exchanges in Europe.

In the following I provide some comments

FIGURES

I think the article would benefit from a supplementary table where the morphometric characteristics of the important populations are summarised. This would help the reader to better follow the taxonomic discussion.

INTRODUCTION

The introduction and the part "the genus Melissiodon" seem to me too long. It would be possible to include the long part corresponding to the family attributions in a remark of the systematic §. Melissiodon is referred by De Bruijn et al. 2013 to Kargi 2 (not 1)

MATERIAL & METHODS

In my opinion, these biochronological proposals should be reworked as a matter of priority. This is important because it is the basis of the discussion. I am not going to discuss everything here but, for example, it seems to me wrong to put Glovelier on the same level as Forsthart. I refer the authors to Kälin & Kempf (2009). A useful figure can also be found in Prieto et al (2018: fig. 9). It also seems unwise to limit oneself to placing Vieux-Collonges in MN 5 without further discussion. I refer the authors to Prieto et al. (2022) and the references therein. What is the argument for placing Béon 2 younger than the Central European sites? I'll leave it at that, but I advise a much more rigorous argumentation.

DISCUSSION

In the course of the discussion, there are numerous references to fossils whose study would shed light on the evolution of the genus (e.g. pp. 509-511). First of all, it is obvious that new data shed more light on the problem. Above all it gives the reader a taste of incompleteness. I would save this for the conclusion in fact. This is found in the § paleoecology.

The authors situate Melissiodon de Kargi at the beginning, but do not return to it. Why not?

Kälin, D. and Kempf, O. 2009. High-resolution stratigraphy from the continental record of the Middle Miocene northern Alpine Foreland Basin of Switzerland. Neues Jahrbuch für Geologie und Paläontologie, Abhandlungen 254 (1/2): 177-235.
Prieto, J., Lu, X.-Y., Maridet, O., Becker, D., Pirkenseer, C., Rauber, G., and Peláez-Campomanes, P. 2018. New data on the Miocene dormouse Simplomys García-Paredes, 2009 from the peri-alpine basins of Switzerland and Germany: palaeodiversity of a rare genus in Central Europe. Palaeobiodiversity and Palaeoenvironments 99 (3): 527-543.
Prieto, J., Rummel, M., Scholz, H., and Mein, P. 2022. A new middle Miocene lineage based on taxonomic revision of the large and rare cricetid-rodent genus Lartetomys. Palaeobiodiversity and Palaeoenvironments 102 (1): 223-236.

---

## Round 0.2 · Minor Revisions

Dear author,

Thank you for the submission of your manuscript entitled “Early Miocene remains of Melissiodon from Mokrá-Quarry (Moravia, Czech Republic) shed light on the evolutionary history of the rare cricetid genus” to PeerJ.

As academic editor I consider that, subject to minor/moderate revisions your paper will be suitable for publication. However it is imperative that a fluent speaker improve the English of your manuscript before its publication. Without the use of proper English throughout your text we will not be able to publish your work. Additionally, I have uploaded an annotated document with some suggestions, so, please, make the appropriate modifications to your paper.

Once again, thank you for submitting your manuscript to PeerJ and I look forward to receiving your revision.

If I can be of any assistance please do not hesitate to contact me.

Best wishes,

Raquel López-Antoñanzas

---

## Round 0.3 · Minor Revisions

Dear author,

It is a pleasure to accept your manuscript entitled “Early Miocene remains of Melissiodon from Mokrá-Quarry (Moravia, Czech Republic) shed light on the evolutionary history of the rare cricetid genus”which you submitted to PeerJ. However, there are some minor changes that you should address before acceptance.

First, you should add a scale with units to Fig. 4.

Second, please, change in the text (p.9, line 144) "...by Carl Zeiss Stemi 305 microscope..." for "...with a Carl Zeiss Stemi 305 microscope..." and in p.9 (line 146) "...by Quanta FEG 250..." for "...with a Quanta..."

Once again, thank you for submitting your manuscript to PeerJ.

Best wishes,

Raquel López-Antoñanzas

---

## Round 0.4 · accepted · Accept

Dear author,

It is a pleasure to accept your manuscript entitled “Early Miocene remains of Melissiodon from Mokrá-Quarry (Moravia, Czech Republic) shed light on the evolutionary history of the rare cricetid genus”which you submitted to PeerJ.

Once again, thank you for submitting your manuscript to PeerJ.

Best wishes,

Raquel López-Antoñanzas